# Exploring the Dynamic of a Circular Ecosystem: A Case Study about Drivers and Barriers

Sophia Barquete [1], Ana Hiromi Shimozono [1], Adriana Hofmann Trevisan [1,*], Camila Gonçalves Castro [1,2], Leonardo Augusto de Vasconcelos Gomes [3] and Janaina Mascarenhas [1]

[1] Department of Production Engineering, São Carlos School of Engineering, University of São Paulo, Av. Trabalhador São Carlense, 400, São Carlos 13566-590, SP, Brazil; sophiabarquete@usp.br (S.B.); anahiromi@usp.br (A.H.S.); camilagcastro@usp.br (C.G.C.); jana.mascarenhas@usp.br (J.M.)

[2] Federal Institute of Education, Science, and Technology of Minas Gerais, Campus Congonhas, Av. Michael Pereira de Souza, 3007, Congonhas 36415-000, MG, Brazil

[3] Faculty of Economics, Administration, and Accounting, University of Sao Paulo, Sao Paulo 05508-010, SP, Brazil; lavgomes@usp.br

\* Correspondence: adrianatrevisan@usp.br

**Abstract:** The circular economy (CE) aims to minimize the environmental impact caused throughout the entire production chain, which can be achieved by implementing circular strategies in collaboration with different actors within a business ecosystem. Although the close relationship between CE and business ecosystem concepts, which originated the term "circular ecosystem", research about this subject is necessary, given the scarcity of empirical studies addressing the phenomenon. Therefore, this study aims to contribute by investigating a Brazilian circular ecosystem specialized in the manufacture of ecological tiles through recycled carton packages. The exploratory case study method was selected to characterize the ecosystem and identify 27 drivers and 17 barriers that enhance and hinder the ecosystem's existence and functioning. Our findings, summarized by a framework, demonstrate the need for integration among the ecosystem's actors so that its value proposition can be delivered. This issue is crucial for collecting post-consumer packaging for recycling and manufacturing ecological tiles. However, actors within the circular ecosystem face some obstacles to collecting the amount of packaging post-consumer material, such as the COVID-19 pandemic. Finally, this work generates discussions and future studies on circular ecosystems, especially in the Brazilian context, where there is little evidence in this research field.

**Keywords:** business ecosystem; circular ecosystem; carton packaging recycling; Brazil

## 1. Introduction

The circular economy (CE) represents a promising attempt to integrate economic activities and environmental well-being into society [1], as it promotes the notion of waste and resource recycling [2]. The transition to this economic model of production is driven by different factors, such as environmental regulation [3], resource scarcity [4], and stakeholder pressure [5], among others. However, companies may face barriers to implementing circular initiatives during their transitional process [6].

Drivers and barriers are important forces to boost the circularity or hamper CE's adoption [3,5]. According to the Oxford Learner's Dictionary [7], a driver is "one of the main things that influence something or cause it to make progress". On the other hand, a barrier is considered "a problem, rule or situation that prevents somebody from doing something". When found in the management literature, these concepts are commonly used to indicate factors that enhance or hinder the desired development [8].

The CE scholars generally examine the drivers and barriers in terms of circular business models (micro-level) and nations (macro-level). For example, Galvão et al. [6] explored barriers and obstacles that can be turned into competitive opportunities by companies.

Agyemang et al. [5] discussed seventeen CE drivers and fifteen barriers applied to the automobile industry in Pakistan. García-Quevedo, Jové-Llopis and Martínez-Ros [9] examined barriers faced by small and medium companies in Europe and suggested that a lack of human resources is critical when companies implement circular business models. Although these studies offer significant theoretical contributions, they do not consider the CE to be meso-level (i.e., ecosystem and industrial parks [10–12]. In particular, the circularity demands changes in the way companies create value and make deals [13], encompassing a systemic perspective with multiple stakeholders' interconnection [14,15]. Consequently, a barrier faced by an actor such as inertia and aversion to change, for example, can impact the evolution of the entire structure in which that specific actor is involved [16].

Overall, companies cannot be seen as isolated entities to achieve circularity [17]. CE involves entire production networks, organized in an ecosystem structure [18], in which responsibilities between actors are diffused [1]. An ecosystem is characterized by heterogeneous and interdependent actors [19] with complementary roles [20,21] and positions [22], who align their activities in order to make a value proposition tangible [22,23]. An ecosystem usually relies on an orchestrator company [24]. This specific actor plays a vital role in the ecosystem, facilitating the communication between members and supporting the creation and sharing of value [25].

Recent studies have started to integrate the concept of an ecosystem with CE, e.g., [26–28]. Aminoff et al. [29], for example, proposed a framework for shaping industrial systems toward circular ecosystems, pointing out that value co-creation from a variety of partners is crucial. In turn, Trevisan et al. [26] presented elements of a circular ecosystem and approached the term as a system of interdependent actors that guides efforts towards a circular value proposition. Hsieh et al. [18] showed how an orchestrator coordinates a glass ecosystem to potentialize circularity. Konietzko et al. [30] provided an example of a multi-actor innovation ecosystem project for a circular economy.

Although these studies offer rich and insightful knowledge on ecosystems in circular contexts, the factors that enhance and restrict this phenomenon's existence and functioning remain unclear. Scholars still call for more empirical research regarding the dynamic of a circular ecosystem and its complex relationships [26,29,30]. We argue that the circular ecosystem might face unique challenges that go beyond the individual barriers of each company's business model. Thus, in this study, we focus on the ecosystem and CE literature to explore a case study of a Brazilian ecosystem specialized in recycling carton packaging to manufacture ecological tiles.

Brazil has more than 212 million inhabitants and generates around 225 tons of solid waste per day in 2020 [31]. This number is quite relevant from the point of view of solid waste since about 1.4% of all solid waste generated in Brazil corresponds to multilayer packages [32]. Even so, there is room to improve the recycling rate in Brazil, and this is a theme of recent policies to develop the solid waste sector until 2040. This also revealed the large importance of research in the country, where the CE and ecosystem theory has the potential to help in the transition.

The National Policy on Solid Waste, established in 2010 in Brazil, contributed to leveraging the CE through collaboration in diverse sectors [33,34] and promotes the shared responsibility of different actors for the proper management of waste [35]. However, the number of studies exploring the CE in Brazil remains scarce [36] even though the subject is receiving greater attention from researchers, industry, entrepreneurs, etc. [37]. The CE is also affecting Brazilian consumers, who are increasingly valuing an eco-friendly market and companies committed to sustainable values [38]. For a country with a large territorial extension, high biodiversity, different cultures and varied socioeconomic conditions, interdisciplinary works that explore circular solutions are needed [39]. Thus, this paper aims to characterize and identify drivers and barriers that boost and limit the circular ecosystem's functioning in the emerging Brazilian economy. Through an exploratory case study, it was possible to understand and analyze the ecological tiles ecosystem from the ecosystem

orchestrator's perspective and other key actors, ensuring a non-exclusive point of view of a single actor.

Our core contribution is a framework that provides a typology of circular ecosystem drivers and barriers and shows the main actors involved, their configuration, and the current changes that have been implemented. Second, we detail the main drivers that boost the circularity within the ecosystem and motivate the set of actors' collaboration. Finally, in contrast to the prior literature, which usually explores the CE barriers on the organizational level, e.g., [5,6,40], we introduce the barriers that impact the circular ecosystem's stability and influence the dynamics of all actors.

This paper is structured as follows: In Section 2, the theoretical background is presented. In Section 3, we detail our methodological procedure. Section 4 offers the results, which are the identification of drivers and barriers. In Section 5, we provide a discussion about our empirical findings, focusing on the emergence of the ecosystem, the cooperation among actors, the competitiveness within the ecosystem and the role of society, and present our theoretical framework. Sections 6 and 7 point out the theoretical contributions and practical implications, respectively. Finally, in Section 8, a conclusion is made, which points out the limitations of our study and possibilities for future work.

## 2. Literature Review

### 2.1. Drivers and Barriers of a Circular Economy

Despite the benefits that the implementation of CE implies for industry owners, customers and the government, ignorance of these benefits results in the neglect of circularity [41]. In addition to this lack of knowledge, many studies describe other barriers and challenges to the implementation of a circular model [41–43]. The lack of initial capital and the difficulty of obtaining investments [44–46], as well as the lack of government support, regarding subsidies and effective legislation [3,45,46], are examples of barriers mentioned in the literature.

In addition to the barriers, the literature identifies, to a lesser extent, opportunities and drivers for the implementation of circular models. In this sense, the government can also act as a driver of CE by implementing effective legislation on waste management, product lifecycle management and laws on hazardous substances, but also investing financially in circular initiatives [3,44]. Other enablers that can be mentioned are the desire to increase the company's prestige, reduce costs and ensure environmental recovery [46].

Despite the existence of numerous studies on barriers and drivers for the implementation of circular models, many are focused on the firm level, discussing, for example, circularity in Small and Medium Enterprises (SMEs) [44–46]. For this type of business or even for startups, it is easier to adopt circular principles, since the company culture is still developing [45]. Circularity, however, demands interaction within an ecosystem of actors, so that the perspective is no longer just about the firm level and moves to the ecosystem level [13].

Regarding the ecosystem literature, the study largely focuses on bottlenecks, defined as components that limit the performance of a system as a whole [47]. Bottlenecks impede ecosystem development and need to be minimized [48]. For example, Hannah and Eisenhardt [49] mention finance as a bottleneck within the US residential solar sector ecosystem, given the high cost of residential solar systems that constrained the industry's growth.

Even though the CE literature already describes challenges and enablers in the context of implementing circular models, works that identify drivers and barriers at the ecosystem level are scarce. Yet, despite studies on bottlenecks in ecosystems [47,50,51], very little is known about the challenges that affect their functioning [52]. Our study, therefore, sheds light on drivers and barriers within the context of a circular ecosystem, contributing to these gaps in the literature.

## 2.2. From a Business Ecosystem to a Circular Ecosystem

The definition of "ecosystem" contemplates a multilateral set of partners, not necessarily hierarchically controlled [20], which interact in order to realize the core value proposition [22]. In other words, the ecosystem encompasses characteristics such as being complementarity [20,53] and actors' interdependence [22] that collaborate and coordinate their activities [20] to build an integrated industrial system focused on business [54].

A business ecosystem can rely on a key company that plays an orchestrator's role which motives and coordinates all the members [24]. Orchestrators are responsible for promoting the ecosystem's health by facilitating communication between stakeholders and by creating and sharing value [25]. In sum, orchestrators must manage different interests and assure alignment among partners [55].

The need to combine circular strategies and business ecosystems to maintain economic and environmental value is clear [18]. The application of the CE in business models ensures competitive advantages, given its potential to create value by means of: (i) its ability to produce at lower costs; (ii) increasing the lifetime of products; (iii) the power of cascading use; and (iv) the power of its own cycles as materials with no contamination can increase material efficiency and productivity [1].

Despite the close relationship between CE and business ecosystems being the subject of many studies, e.g., [14,17,26], the concept of "circular ecosystems" itself is still recent and is being increasingly studied. Empirical studies are still needed to demonstrate the phenomenon and identify its characteristics, drivers, and the rebound effects of the transition to the circular ecosystem [56].

## 3. Materials and Methods

In this paper, the subject of analysis is an ecosystem of ecological tiles originating from recycled carton packages. A single case study method was adopted for two main reasons. First, this method allows us to understand a decision-making series (e.g., what, how and why). Second, it allows the investigation of a real-life phenomenon whose full extent is unknown [57]. This method was critical to answering questions such as "what are the drivers that boost the ecosystem's development?" and "what are the main barriers the actors face within the ecosystem?".

In order to characterize the studied ecosystem, the 6C framework proposed by Rong et al. [58] was adopted. The framework allows a broad understanding of an ecosystem by describing it in six spheres: context, construct, configuration, cooperation, capability and change. Furthermore, given that ecosystems consist of different actors [59], several companies were contacted to avoid an impaired comprehension due to a biased view of one interviewee which could jeopardize the study's outcome.

### 3.1. Data Collection

As proposed by Eisenhardt and Graebner [60], a case study collected data from different sources. The main data collection methods used in this paper were: interviews with CEO, managers and specialists, informal conversations through e-mails and messages, in order to clear occasional doubts, and a single site visit. An additional search was carried out on the official websites of the studied companies, as well as social media and news websites.

To assist the researchers in conducting interviews, a protocol was elaborated as it is one of the essential tools when conducting several interviews since it assures that the main topics will be covered. Furthermore, it anticipates possible problems that might arise during the interviews [57]. In this protocol, the questions addressed the following topics: the company's history, the ecosystem's actors and the relationship between them, company perceptions about CE, the CE practices adopted and the main challenges that were faced.

The primary data came from interviews with 5 different companies with essential and indispensable roles within the ecological tiles ecosystem. We employed the snowballing technique [61] to select the key actors within the ecosystem. Interviews began

with the carton packaging manufacturer (the orchestrator) that indicated other players to be interviewed.

From this first contact with the ecosystem's orchestrator, it was possible to identify and map the main necessary partnerships to the functioning of the ecosystem. Therefore, in addition to the orchestrator (interviews 1 and 2), 4 other companies were interviewed: two ecological tiles manufacturers (interviews 3–5); the company responsible for recycling and transforming the waste in another product (interview 6); and finally, the waste management company responsible for promoting the waste value within the ecosystem (interview 7). The interviews are identified in Table 1.

**Table 1.** Interview list.

| Interviewee's Identification | Company | Interviewee's Position | Method |
|---|---|---|---|
| Interviewee 1 | Carton Packaging Manufacturer–Orchestrator | Sustainability Manager | Sustainability report and interview |
| Interviewee 2 | Carton Packaging Manufacturer–Orchestrator | Sustainability Manager | Sustainability report and interview |
| Interviewee 3 | Ecological Tiles Manufacturer–1 | Specialist | Interview and site visit |
| Interviewee 4 | Ecological Tiles Manufacturer–2 | Administrative Manager | Interview |
| Interviewee 5 | Ecological Tiles Manufacturer–2 | CEO | Interview |
| Interviewee 6 | Recycling Company | Business Specialist | Interview and secondary data (website) |
| Interviewee 7 | Waste Management Company | Marketing Manager | Interview and secondary data (website) |

Interviews were conducted between March and November 2021, all carried out virtually with an average duration of 60 min each. In December 2021, following all COVID-19 security protocols, a site visit was conducted to the Ecological Tiles Manufacturer 1 to eliminate outstanding doubts about the tile production process. For the interviewees to be able to express themselves openly and consequently provide a more in-depth notion about their daily issues, semi-structured interviews were implemented, and all participants were instructed about information anonymity. Additionally, a consent term was signed by all interviewees to ensure everyone agreed with the project objectives. With their approvals, the conversations were recorded and transcribed, so the content lost would be as minimal as possible.

### 3.2. Data Analysis

As multiple data sources were used (interviews, observations and reports), the analysis began with data triangulation to validate the collected data and assure a deeper comprehension [62]. With data being collected through interviews and annual sustainability reports, it was possible to understand the main event's evolution, such as the establishment of key partnerships.

Afterward, a coding process of the interviews was carried out through MAXQDA® [63] software mainly applied to qualitative data analysis. Following the methodology proposed by Miles, Huberman and Saldaña [64], two coding cycles were performed for both drivers and barriers. Figure 1 illustrates the coding process.

The first cycle aims to perform a first "scan" and obtain, in a rough way, descriptive codes about the drivers and barriers. According to Gioia, Corley and Hamilton [65], a second cycle is necessary to review the codes one by one, performing a more careful analysis. This process allowed us to group a series of codes according to their similarities and eliminate mistakes made during the first cycle. As part of a rigorous coding process, the two mentioned cycles were analyzed and verified by two people and, once categorized, the codes were further discussed with a third author.

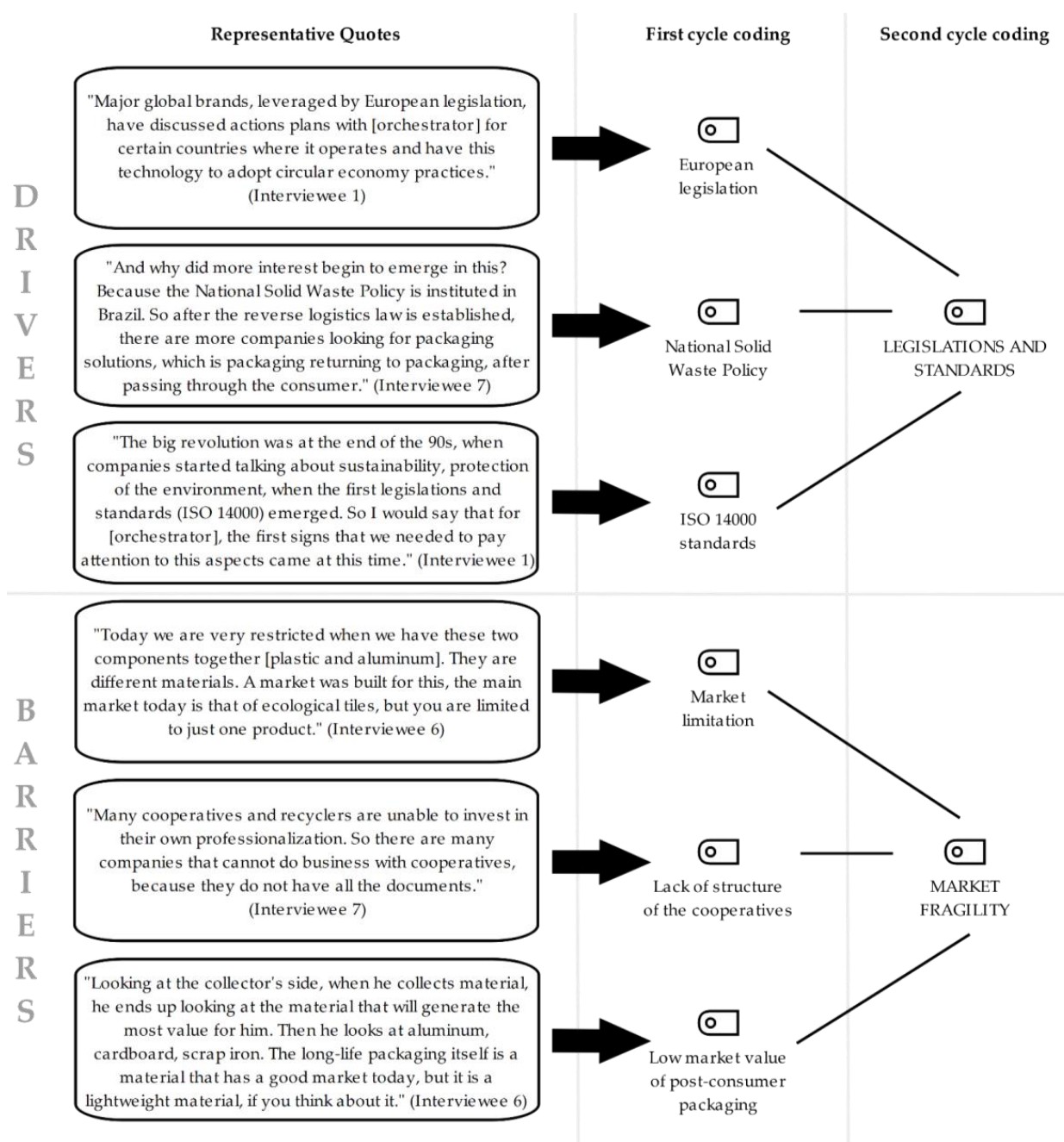

**Figure 1.** Example of data-structure tree regarding the drivers in the "Legislations and standards" category and the barriers in the "Market fragility" category.

### 3.3. Case Description—Ecological Tiles Circular Ecosystem Characterization

The 6C framework developed by Rong et al. [58] was used to characterize the circular ecosystem of this work. The framework considers six spheres as necessary for the description of a business ecosystem: context, cooperation, construct, configuration, capability, and change [58]. These elements of the ecological tiles are presented in this section and summarized in Table 2.

**Table 2.** Ecological tiles circular ecosystem characterization based on the 6C framework of Rong et al. [59].

| 6C Element | Description | Characterization |
|---|---|---|
| Context | The basis for the network to develop | Boom in the civil construction after 2007 and the subsequent Brazilian legislation on solid waste (National Policy on Solid Waste-PNRS). The mission of the ecosystem is to recycle the carton package and prevent it to go to the landfill. |
| Construct | The skeleton of the network | Legislation, carton packaging manufacturers, food industries, collectors and recyclers, ecological tiles manufacturers, sellers, NGOs, waste management companies; development of technology to process the carton packaging; traceability of post-consumer carton packs. |
| Cooperation | Mechanisms behind partners interactions | The orchestrator started to establish different partnerships to promote the recycling of its product. Both contractual and non-contractual were found. |
| Configuration | The way the construct elements are combined | The orchestrator created the ecosystem and is still driving its development. They fund technology for partners and share the responsibility of attracting more players into the ecosystem with the tiles manufacturers. |
| Capability | The reflection of the configuration | There is a vicious circle of the orchestrator pushing their partners intellectually and financially; The partners are able to collect and process more material; the orchestrator and the food industries can correctly dispose of the waste, complying with legislation. The orchestrator also developed technologies for partners in the ecosystem. |
| Change | The necessary changes in order to evolve—renewal, co-evolution | Existing partnerships evolved to supply other materials for product diversification due to the orchestrator's low supply of carton packs. Investment in product development with the market downturn due to COVID 19. |

The circular ecosystem of ecological tiles originated from the recycling of carton packaging and emerged from the possibility of recycling the remaining material from this industry. This process was accentuated by the boom in the civil construction market between 2007 and 2008 in Brazil. The National Policy on Solid Waste (PNRS), regulated by Brazilian legislation in 2010, also contributed to the participation of actors in the ecosystem, as it obligates waste generators to correctly destinate their waste, evidencing the need for companies to implement reverse logistics [66]. The ecological tile and sheet market represented about 90% of the plastic-aluminum recycling market with recycling incentives. In addition to these data, 44% of the carton packages produced in 2020 by the carton packaging manufacturer (the orchestrator of the ecosystem) are recycled.

Besides the orchestrator company, which holds 80% of the market share in Brazil regarding packaging production, the ecosystem has several heterogeneous actors essential for the ecosystem's output (ecological tile) to materialize. Figure 2 shows the main stakeholders of the ecosystem, grouped according to their areas of activity.

Besides the structure of actors, other resources enable the ecosystem to reach its potential. The equipment initially developed by the orchestrator and loaned or sold to partners is essential for the processes of recycling carton packaging and manufacturing ecological tiles. Furthermore, the partnership signed between the orchestrator and the technology company, based on blockchain to promote the traceability of post-consumer carton packs, guarantees reverse logistics credits in cash to those who prove the correct collection and disposal of carton packaging. Figure 3 shows the configuration of the ecological tile ecosystem with material and data flow.

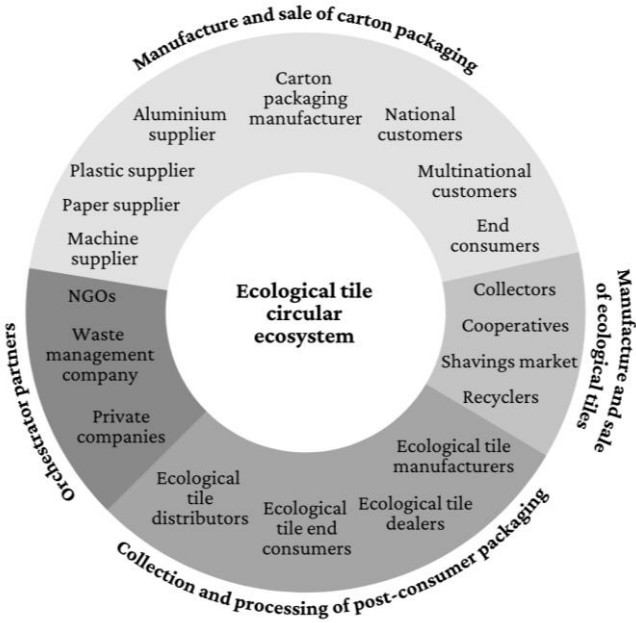

**Figure 2.** Actors of the ecological tiles circular ecosystem.

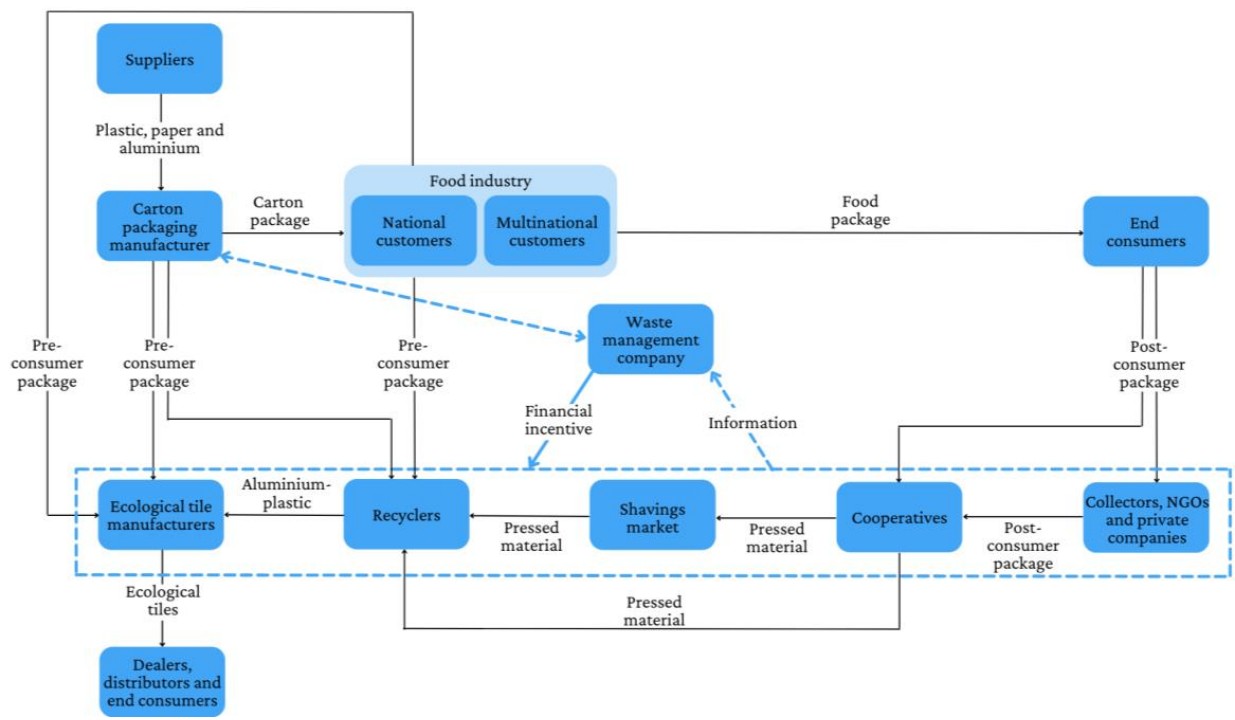

**Figure 3.** Material and information flow among actors in the ecological tiles ecosystem.

Both contractual and non-contractual relationships characterized these partnerships. An example of a contractual relationship was the establishment of lending contracts (free loan on condition of return after a certain period) of equipment for recycling. Regarding non-contractual relationships, many collectors, cooperatives, waste pickers, recyclers, non-governmental organizations (NGOs), private companies, or even manufacturers of products that use the waste from carton packaging as input for their processes act on their own in the collection and treatment of this material. In addition to ecological tiles, some other products are produced from post-consumer packaging, which drives the demand for this material.

The production process of ecological tiles starts with supplying raw material to the ecosystem orchestrator to manufacture carton packaging, which is sold to national or multinational food industries. After the product retail and reach the final consumers, the post-consumer packaging arrives at the cooperatives through collectors or through the selective collection system in the case of correct disposal by the final consumers. Then, the cooperatives separate this material and sell it pressed to shavings markets or directly to recyclers. These, in turn, separate the elements from the carton packaging and supply the clean aluminum-plastic to the ecological tiles manufacturers. They then reinsert the material on the market in the form of a final product.

Besides post-consumer packaging, the pre-consumer material, cleaner than post-consumer packaging, goes directly to the recycling company or even to the manufacturers of ecological tiles when they have equipment for separating the aluminum-plastic. Given the need to boost the recycling of generated waste, the orchestrator established a partnership with the waste management company, which works by connecting the orchestrator with those who correctly dispose of their carton packaging. By proving, through invoices, the amount of packaging that is being correctly destined, this company pays collectors, cooperatives, shavings markets, recyclers, the ecological tile manufacturers themselves, and everyone who buys and sells their post-consumer material through recycling credits.

Within a network, the individual capabilities of the actors are enhanced [24]. The orchestrator can help its partners with equipment and processes know-how by actively seeking solutions promoting reverse logistics and improving the ecosystem. Influenced by the orchestrator, all those who collect post-consumer carton packaging can provide a considerable amount of material to be recycled because these actors can be paid additionally for the collection and correct disposal of the carton packaging produced by the orchestrator. Therefore, recyclers and manufacturers of ecological tiles have greater chances of getting waste material to make their processes work within the ecosystem. At the same time, in addition to the orchestrator, the food industries can correctly dispose of residual pre-consumer packaging, complying with Brazilian legislation.

Before the COVID-19 pandemic, the demand and supply of the ecological tiles ecosystem were balanced. The demand for ecological tiles remained high and stable, and the supply of residual material from packaging, pre- or post-consumer, for the manufacture of tiles was large and sufficient. However, the pandemic in Brazil was followed by a drastic reduction in the supply of post-consumer packaging for recycling, as will be explained in detail in Section 4.1.2. The need to adapt to this new reality forced the different actors to look for alternatives inside and outside the ecosystem. The tile manufacturers, for example, began to study the feasibility of using other materials, usually discarded in landfills, to manufacture tiles and develop new or more profitable products, such as transport angles and pallets. Furthermore, one of the manufacturers invested in its own company to collect and distribute pre-consumer material, aiming to overcome the limited and expensive supply of this material. During this pandemic period, the partnership between the orchestrator and the waste management company was signed to boost the collection and recycling of post-consumer packaging.

## 4. Results

### 4.1. Drivers and Barriers in a Circular Ecosystem

This section presents the 27 drivers and 17 barriers that enhance and hinder the ecosystem's existence and functioning. The findings revealed that a circular ecosystem can be driven by the contextual factors of ecosystem birth, legislation, market pressure, the collaboration between actors, and the material properties of the circular product. In addition, the circularity within the ecosystem may be hindered by the lack of material, high cost, market fragility, poor alignment, and manufacturing problems.

### 4.1.1. Drivers

The drivers for the development and functioning of the ecological tiles ecosystem are shown in Table 3. The findings are grouped into five categories that exemplify the main forces that enhance the circular ecosystem evolution.

**Table 3.** Drivers of a circular ecosystem.

| Category | Drivers | Description |
|---|---|---|
| Birth context of the ecosystem | Emergence of carton packages (D1) | The emergence of the orchestrator company, a large manufacturer of carton packaging, influenced the search for recycling its waste. |
| | Emergence of processes and equipment for carton packaging recycling and ecological tiles manufacturing (D2) | Development of processes and equipment for both the recycling of carton packaging and the manufacture of ecological tiles for the orchestrator. |
| | Heated civil construction market (D3) | The booming civil construction market between 2007 and 2008, with a large number of houses being built, influenced the development and commercialization of ecological tiles due to the recycling of carton packaging. |
| Legislations and standards | National Policy on Solid Waste (D4) | The policy imposes mandatory collection and recycling of products and their waste after consumption by the end customer, by manufacturers, importers, distributors and traders of these products. |
| | European legislations (D5) | Europe, the continent where the orchestrating company comes from, influences Brazilian subsidiaries in adopting circular economy practices in accordance with its laws. |
| | ISO 14000 standards (D6) | ISO 14000 consists of a serie of standards that establish guidelines to ensure that a particular company (public or private) practices environmental management. |
| Pressure from environmental issues | Environmental impacts (D7) | Environmental impacts, such as climate change, influence the adoption of a more responsible production chain, concerned with the generation of waste and pollutants. |
| | Internal goals related to sustainability (D8) | Orchestrator's internal goals to promote the circular economy through the manufacture of more sustainable carton packs and the support for selective collection and recycling strategies. |
| | Pressure for the adoption of sustainable solutions (D9) | Pressure from the carton manufacturer's customers and end consumers to adopt sustainable measures with regard to waste carton packs. |
| | Demand for green products (D10) | Demand for greener and more sustainable products by society. |
| | Impossibility of discarding carton packages in landfills (D11) | Given the large scale of carton packaging production, it is necessary to think about reverse logistics solutions within the ecosystem, rather than just disposing of waste in landfills. |
| | Easily disposal provided by the ecosystem for the correct destination of pre- and post-consumer packages (D12) | The ecosystem provides an accessible solution for the correct disposal of carton packaging waste, both pre- and post-consumer, offering a facility for actors that generate this waste and, consequently, encouraging their participation. |
| Cooperation among actors | Incentive for recycling carton packages and producing ecological tiles (D13) | After developing processes and equipment for recycling carton packs and manufacturing ecological tiles, the orchestrator sought partners to put these processes into practice, providing technical and financial assistance. |
| | Loan and sale of equipment (D14) | Orchestrator lends or sells equipment to partners willing to carry out the processes of recycling carton packaging or manufacturing ecological tiles. |
| | Incentive for the active participation of end consumers (D15) | Encouragement by ecosystem actors so that end consumers dispose of post-consumer packaging correctly to promote the collection and recycling of material. |
| | Establishment of post-consumer packages recycling stations (D16) | Assistance on how to deal with post-consumer packaging and where to dispose of them, given mainly by the orchestrator, as the manufacturer of the carton packs, through its own application that shows the location of the various recycling stations. |
| | Financial incentive for selective collection (D17) | The orchestrating company financially rewards partners who correctly collect and dispose of their waste. |
| Properties of carton packages and ecological tiles | Design for environment of carton packages (D18) | Constant readjustments are made in the design and composition of the carton packages, in order to facilitate the collection and recycling of the material. |
| | Differences of ecological tiles in relation to other tiles options (D19) | Ecological tiles are durable and guarantee thermal comfort, which drives its choice among less sustainable ones and guarantees demand for what the ecosystem generates as an output. |
| | Possibility of cyclic recycling of ecological tiles (D20) | Eco-friendly tiles can be recycled cyclically, which means that tiles used after years or those manufactured with defects can be used as input for the manufacture of new tiles. |

The first drivers' category is called the "Birth context of the ecosystem" and encompasses the drivers related to initial forces that fostered the circular ecosystem's development. Due to the "Emergence of carton packages" (D1), the possibility of exploring new market niches, such as recycling discarded material and manufacturing products from recycled material, was established. In this context, studies conducted mainly by the orchestrator

were essential for the "Emergence of processes and equipment for carton packaging recycling and ecological tiles manufacturing" (D2). As stated by interviewee 1: "*in 2005, the first plastic-aluminum recyclers began to appear [ . . . ]. We started to develop plastic-aluminum recycling processes, which began with pellets, those little plastic spheres that are very common in the plastics industry [ . . . ], and then, in the 2007, through ecological tiles and sheets that today are practically 90 % of that aluminum-plastic market. We helped develop the thermoforming process, already used in the timber industry, of heating this material, making a malleable sheet and molding it into the shape of a tile or using it as a sheet that replaces timber. This process was done by combining technology, availability of investment and, of course, market development of the final product*". A third driver related to the ecosystem's birth context is that, between 2007 and 2008, the "Heated civil construction market" (D3) allowed ecological tiles to gain space as a commercialized product. During this period, a "*houses boom*" occurred, as mentioned by interviewee 1, resulting in a very well-developed market in the civil construction sector, which encouraged the production of ecological tiles from carton packaging recycled as a viable product with great demand.

The second drivers' category refers to bureaucratic issues, such as imposed legislation and rules. As the first driver in this category, the "National Policy on Solid Waste" (D4) from 2010 forced companies to correctly dispose of their waste, as emphasized by interviewee 4: "*Today, companies want to get involved with the environmental part of the business, because they need to, they are obliged to give the correct final destination to what they generate*". In this context, the orchestrator closed a partnership with the waste management company to boost post-consumer packaging collection. According to interviewee 7, from the waste management company, "*they [orchestrator] came to us, because they wanted to encourage the chain for which they are responsible. They take responsibility for what they put on the market.*" As the second and third drivers of this category, our data suggest the "European legislations" (D5) and the "ISO 14000 standards" (D6). Europe, the home continent of the orchestrating company, presses through its laws the adoption of circular economy practices, as stated by interviewee 1: "*the great global brands, leveraged by the European legislation, have discussed with the [orchestrator] action plans for certain countries where it operates and has this technology to adopt circular economy practices*". Furthermore, the ISO 14000 standards, which establish criteria for environmental management within companies and organizations [67], encourage discussions on this subject and the adoption of sustainable practices.

The third category, "Pressure from environmental issues", relates to environmental degradation and the need to adopt sustainable solutions. "Environmental impacts" (D7) are the first driver of this category. Regarding the incorrect disposal of solid waste, reducing pollution motivates actors to seek less polluting alternatives for their businesses, including establishing "Internal goals related to sustainability" (D8), which guide the actors' principles toward the CE. In addition to that, there is also the "Pressure for the adoption of sustainable solutions" (D9) by ecosystem customers (e.g., food industry brands and end consumers) for the correct disposal of packaging waste and the "Demand for green products" (D10), evidenced by interviewee 3: "*in the past, nobody used to buy [ecological] tile thinking 'it's a recycled tile, like 'I'm going to help the environment. Nobody was like that. But now, for some time now, I have noticed that people have changed the concept a little: 'let's look for an ecological tile'*". Due to the dimension of the carton packaging production headed by the orchestrator, there is the "Impossibility of discarding carton packages in landfills" (D11) as another driver. According to interviewee 1, "*at the time [the 2000s], they took this material [plastic-aluminum] and sent it to the landfill. This, in the 2000s, when the [orchestrator] produced 1 billion packages, was not a significant amount. But today, there are already 11 billion [ . . . ]. Today, we generate around 20,000 tons of plastic-aluminum, so we can't throw it all in a landfill*". From then on, the recycling processes were developed until reaching the current configuration of the ecosystem. In the current configuration, the "Easily disposal provided by the ecosystem for the correct destination of pre- and post-consumer packages" (D12) motivates the actors within the circular ecosystem. Mainly for pre-consumer packages,

actors in the food industry are exposed to the possibility of quickly discarding defective packaging and fulfilling their responsibility for the correct disposal of waste.

The fourth category, called "Cooperation among actors", relates to interactions between ecosystem stakeholders and how they help each other in order to potentialize the joint development of the desired solution. As the first driver in this category, we have the "Incentive for recycling carton packages and producing ecological tiles" (D13) by the ecosystem orchestrator. After studying and developing the processes needed, the orchestrator searched for possible actors that could implement and carry out the recycling of post-consumer packaging and the production of ecological tiles. Interviewee 3, a specialist in ecological tiles manufacturing, shows this demand on the part of the orchestrator: "*In the beginning, it was an idea of the [orchestrator] itself, they said: 'we are going to make an ecological tile with our material', we agreed, and we were perfecting the process. Today it is one of our flagships*". In addition to intellectual support, the orchestrator also provided technical support to these actors through the "Loan and sale of equipment" (D14), which facilitated the implementation of the desired processes. "*We basically had this partnership: you set up your line, maintenance costs, equipment, people, all of this is yours within your plant, and the [orchestrator] can lend 2, 3 or 4 pieces of equipment to your line works. This model worked very well, and today we have many recyclers who come to us to invest by themselves, as they have already seen in other plants that the product is good and economically worthwhile*", explains interviewee 1 concerning the equipment lending.

Another issue that demands cooperation among the actors is selective waste collection. In this sense, the "Incentive for the active participation of end consumers" (D15) and the "Establishment of post-consumer packages recycling stations" (D16) are drivers that support the society with the means to the correct destination of post-consumer waste. For this, the orchestrator developed its own mobile application, in which actors, including end consumers, can register and receive information about the carton packaging, how to work with it and, above all, how to dispose of it correctly. Furthermore, the application determines collection locations, as interviewee 1 explains: "*it is a place to centralize the collection locations for carton packaging in Brazil [ . . . ]. These locations already exist, and we [the orchestrator] simply mapped them to keep them in a database available for the consumer to know where he can dispose of his recyclable material*".

Another driver that strengthens the ecosystem is the "Financial incentive for selective collection" (D17). According to this driver, the orchestrating company rewards the partners who correctly collect and dispose of their waste by giving them money. An example of how this is carried out is related to the project in conjunction with the waste management company. The project makes use of blockchain technology to enable post-consumer packaging traceability so that the orchestrator is able to financially compensate the actors who prove the selective collection of their packaging. This issue is explained by the representative of the waste management company (interviewee 7): "*We are building, even finalizing, a platform for their operators [orchestrator's operators], partners who buy and sell their material, register the invoices and with that, for each mass, volume invoice, they [orchestrator] pay an additional. It's a way for them [partners] to earn a little more money. Since the market does not value it so much, they [the orchestrator] themselves value recycling the product*".

The fifth and final driver category identified was "Properties of carton packages and ecological tiles", related to the characteristics of the finished products that circulate through the ecosystem (i.e., carton packaging and ecological tiles). First, in agreement with what interviewee 1 says, "*it is necessary to have a design for environment, to think even about the composition of a packaging so that it can be really recycled*", a first driver of this category is related to the "Design for environment of carton packages" (D18). The two other drivers are related to ecological tiles: "Differences of ecological tiles in relation to other tiles options" (D19) and "Possibility of cyclic recycling of ecological tiles" (D20). The first refers to the long duration of ecological tiles and the thermal and acoustic comfort they generate, which allows them to be chosen among other less sustainable tile options and guarantees their demand. The second shows the possibility of cyclic recycling of the tiles, either after years

of use or in case of a defect or leftover material in the factory. Interviewee 4 highlights this property: "*they [ecological tiles] are 100% recyclable. So much so that the manufacturing process does not generate any kind of waste: everything that is made there, its squaring, the part that is left over goes back to production, it is redone*". In conclusion, the properties of the circular product, enabling the continuous recycling of the tiles, are drivers that help maintain the circular ecosystem's proposition.

### 4.1.2. Barriers

The barriers regarding the circular ecosystem development were classified into five categories, namely lack of material, high cost, market fragility, poor alignment between actors, and manufacturing problems (Table 4). A description of each category is presented as follows:

**Table 4.** Barriers to a circular ecosystem.

| Category | Barriers | Description |
| --- | --- | --- |
| Lack of material | COVID-19 pandemic (B1) | Due to the pandemic, many collectors were unable to collect waste, resulting in a decreased volume of recycled cartons. |
| | Replacement of carton packages (B2) | In the food industry, carton packages are being replaced by other packages (e.g., plastics). |
| | Materials that do not meet the specifications (B3) | Often, materials are declined because they do not meet the specifications required by the company (e.g., humidity). |
| | Low availability of pre-consumer packages (B4) | One of the main sources of material acquisition are the pre-consumer packaging, discarded by industries. With the investment in improvements for the reduction of operational costs, the industrial disposal is getting smaller. |
| | Difficulty of small collection (B5) | Recycling companies do not take advantages of small collection, as such work requires high cost of transportation and available labor. |
| High cost | High cost of material transportation (B6) | The high cost of transportation makes it difficult to collect materials in regions far from where the recycling company is located. |
| | High cost of cleaning the post-consumer packages (B7) | The use of post-consumer packaging is little encouraged due to the need to clean the material, increasing the operational cost. |
| | High cost for digital transformation (B8) | Software purchasing and production line changes require large investments by companies, making the transition to the digital world difficult. |
| | High cost for separating plastic and aluminum (B9) | Despite the encouragement of research for the separation of aluminum and plastic in carton packages, this process has a high cost, making it unattractive to recycling companies. |
| Market fragility | Market limitation (B10) | The market for recycling carton packages is limited to the manufacture of ecological roof tiles, with few applications. |
| | Lack of structure in cooperatives (B11) | Many cooperatives are not yet as well structured and professionalized as the private market, which makes it difficult to trade with them. |
| | Low marketing value of post-consumer packages (B12) | Due to the higher sales value, collectors prefer to collect other materials (e.g., cardboard, iron) rather than carton package. |
| Poor alignment among actors | Lack of consumer awareness (B13) | Many consumers do not dispose of the packaging correctly, making it impossible to recycle and reinsert it in new products. |
| | Lack of awareness of companies within the ecosystem (B14) | The concept of sustainability is not yet widely adopted by companies. |
| | Lack of inspection (B15) | Despite the existence of legislation, the enforcement of environmental laws is still very mild by the government. |
| Difficulties in the manufacture of ecological tiles | Manual manufacturing process (B16) | The production volume of ecological tiles is lower, if compared to fiber cement tiles, because it is still a very artisanal process. |
| | Lack of labor (B17) | Due to the manual process, the demand for labor is high, but difficult to find in the regions near the manufacturing companies. |

The first category, "Lack of material" refers to barriers that hinder the purchasing of essential materials to manufacture ecological tiles, one of the categories with the most significant impact on the ecosystem's circularity. The "COVID-19 pandemic" (B1) was one of the barriers that most affected the volume of carton packages collected since different decrees adopted by each Brazilian state prevented many collectors from going out into the streets and collecting material. Another factor worth mentioning, as quoted by interviewee 4, was: "*many people who counted on this [selective collection] started receiving emergency aid, which ended up stopping the recycling gear*". The emergency aid was a temporary Brazilian government program, created in April 2020, which provided a minimum income to informal workers and low-income families [68] whose incomes were affected by the COVID-19

pandemic. The emergency aid and the restrictions imposed by COVID-19 were important to mitigate the damage caused to Brazilians in vulnerable situations, and the impact caused on recycling during this period was very significant.

The "Replacement of carton packages" (B2) is another barrier that has been noticed recently. According to interviewee 4: "*before, we used to see carton packages and now we notice that those packages are being replaced by plastic ones, for instance, so carton packages manufacturers are losing a little space in the food industry*". This replacement ends up directly affecting the discarded material volume and, consequently, the collected waste volume. Among the materials collected, there are also those "Materials that do not meet the specifications" (B3), which present, for example, high levels of humidity or paper that make it impossible to use them for tile manufacturing.

Another source of the material is pre-consumer packaging. The challenge that ecological tile manufacturers are having to deal with at the moment is the "Low availability of pre-consumer packages" (B4), since, as mentioned by interviewee 4, food industries are increasingly committed to "*reducing factory operating costs, or even reusing the material themselves rather than discarding it*". Although such a reduction is extremely beneficial to the ecosystem's circularity, since it aims to reduce waste generation and, consequently, the environmental impact, it also reinforces the difficulty of ecological tile manufacturers to obtain material for production.

The last barrier in this category concerns the "Difficulty of small collection" (B5). Although there are several recycling stations for carton packaging, which play an important role within the ecosystem by promoting sustainability and helping to reduce impacts, the collected volume is not so expressive and does not meet the high demand of tile manufacturers. Recycling companies do not take advantage of these small initiatives as much, mainly because "*we don't have the necessary labor, so we respect the chain as the way it is*" as interviewee 6 said, as well as because of the difficulty in logistic terms of connecting the small recycling stations.

The second category was "High cost", in which the most significant costs that hinder the functioning of the ecosystem were addressed. The first barrier within this category is the "High cost of material transportation" (B6). Although different places offer recycled material, such as the small recycling stations mentioned above, due to Brazil's continental dimension, "*there are some places that make it impractical to do business*", as stated by interviewee 3, mainly because "*waste transportation in Brazil is very difficult and very expensive*", adds interviewee 7. Besides this difficulty in transporting waste, the interviewees mentioned that the "High cost of cleaning the post-consumer packages" (B7) is another factor that leads them to prioritize the use of pre-consumer packaging over post-consumer ones. As companies need to ensure that the final product does not include "*any type of residue for the manufacture of tiles*" (interviewee 4), this cleaning is essential, but due to its high cost, the use of such material ends up not being so attractive to manufacturers.

Another factor that proved to be a barrier was the "High cost for digital transformation" (B8). The need for digital transformation became even more evident with the pandemic since many companies were forced to use new technologies to adapt to the social distance's reality. Despite the advantages they bring, such as the waste tracking carried out by the valuation company, which was only made possible by using blockchain technology, these technologies often come with a high cost, which hinders their adoption. The last barrier in this category is the "High cost for separating plastic and aluminum" (B9) in carton packaging. Despite the orchestrator's incentives to search for new technologies to promote this separation, according to interviewee 6, "*the energy cost did not close the bill*". This process, which would be important for recycled product manufacturing only with plastic or only with aluminum, ends up being financially unviable for the recycling companies.

In the third category, "Market fragility", the data suggest that the barrier "Market limitation" (B10) is directly related to the "High cost of separating plastic and aluminum" (B9). Due to the economic impossibility of separating these two components, the market built from the carton packaging recycling was "*limited to a single product, the ecological tile*",

as interviewee 6 said. Because of this limitation, if the ecological tile market suffers any kind of instability or setback, today the plastic-aluminum resulting from recycling carton packaging would not have any other application.

The "Lack of structure in cooperatives" (B11) was also found to be a market fragility, since, according to interviewee 7, "*the professionalization margin is still very low in this sector*". The biggest problem with this high level of informality is that many cooperatives still do not have all the documents regularized and, consequently, many are not able to close business deals with other companies. Ensuring the professionalization and regulation of these cooperatives is essential. Firstly, they must receive fair value for their waste and secondly, the ecological tiles manufacturers must be able to purchase the necessary material.

There is also a barrier regarding the "Low marketing value of post-consumer packages" (B12). As in Brazil, the waste sales are usually made based on weight (kg), the collectors give priority to materials that have a higher sales value per kilo, which is explained in interviewee 6's speech: "*[the collector] ends up looking for the material that generates more value for him. So they look for aluminum, cardboard, and scrap iron. Carton packaging itself is a material that today has a good market, but it's a light material*". It is also worth mentioning that despite the financial incentives offered by the orchestrator, as presented in driver (D17), many collectors are still informal, which hinders their access to those incentives.

The fourth category "Poor alignment among actors" addresses some failures present in the relationship and activities alignment among the players of this ecosystem, the first of them being the "Lack of consumer awareness" (B13). These players have a key role in the circular ecosystem, which is the correct disposal of packaging after consumption. Despite the instructions present in the packaging itself on how to dispose of it, many consumers still discard their packaging in common garbage dumps, which makes the correct collection of this waste impossible. This lack of environmental awareness was evident in a branding study conducted by the recycling company, which showed that most respondents were unaware that carton packaging is recyclable. According to interviewee 1, "*It's still difficult to convince people that they need to do their part*", mainly because "*we [Brazil] have a problem with basic education [ . . . ], so how can we demand formal education from someone who doesn't have informal education, doesn't have access to the system, is struggling to survive and still we're asking them to do selective collection?*".

Besides consumers, there is a "Lack of awareness of companies within the ecosystem" (B14). Despite being part of a circular ecosystem, many companies still find it difficult to adopt a sustainable culture and practices, as evidenced in the speech of interviewee 6: "*within companies, the basis is still not sustainability, few are [sustainable]*". It can be seen that there is still a difficulty in understanding sustainability as an economic opportunity, besides the importance of responsibility to the environment. This lack of action by companies is partially related to the barrier "Lack of inspection" (B15). Despite the presence of important legislation for the regularization of sustainable practices, due to the lack of inspection, many companies prefer to take risks of incurring small fines compared to the profits obtained rather than adopt such practices.

The last category covers the "Difficulties in ecological tiles manufacturing" faced by tile manufacturers, the "Manual manufacturing process" (B16) being one of them. As mentioned by interviewee 3, "*It is a very handmade process [ . . . ], there is no machine to make this type of tile*". Besides this difficulty, companies need to deal with external factors, for example, "*When it's a very intense rainy season, the material comes with a lot of humidity, this hinders the production. There are times of the year when I have 50% humidity in the process, this is very troublesome, there is a lot of corrective maintenance. Because of the characteristics of the process, the productivity level of ecological tile manufacturers is still very low and "does not reach 5%" of the total tiles produced in Brazil*", according to interviewee 4.

Finally, the last barrier identified was the "Lack of labor" (B17). Interviewee 4 emphasized during the conversation that "*( . . . ) today labor is quite complicated in our region, in general, actually*", which is one of the biggest challenges faced at the moment. Still, as

previously discussed concerning the barrier "High cost for digital transformation", factory automatization is a difficult reality to be achieved.

## 5. Discussion

In this section, five main points related to the ecological tiles circular ecosystem are discussed. Firstly, the reasons behind the emergence of the circular ecosystem and the motivations of actors to participate in the network, which are identified by several of the drivers described. Then, we will address the issue of cooperation among actors, identified as a key category for both drivers and barriers. As a third topic, the difficulty of obtaining pre-and post-consumer material for the manufacture of ecological tiles and the need for ecosystem actors to adapt to this reality will be discussed. Fourth, competitiveness in the tile sector will be highlighted. Finally, the role of society, as a final consumer, will be discussed.

While the drivers of the category "Legislations and standards", D4, D5, and D6, point to the existence of political pressure, the drivers of the category "Pressure arising from environmental issues", D7, D9, D10, and D11, refer to external factors related to the environment that impose on the actors the need to adopt sustainable solutions. In any case, the point is that the birth and development of the ecological tiles' circular ecosystem is primarily due to external, political, corporate, environmental or even social pressures, and not the benefits provided by a circular economy. As much as several studies, e.g., [15,69,70] suggest that economic gains are factors that directly drive the adoption of more sustainable practices, in the case of this particular ecosystem, external pressures outweigh the financial aspects.

In addition to the motivations behind the birth of the circular ecosystem, cooperation is another key point when it comes to circularity. At the same time that cooperation among actors within an ecosystem is essential to foster the value proposition, given that a single company would hardly be able to hold all the necessary technology and know-how, this type of relationship is difficult to be implemented. In the context of the ecosystem studied, a proof of that is the identification of categories related to cooperation among actors for both drivers and barriers.

The drivers that mention cooperation, D13, D14, D15, D16 and D17, refer to the entire process of correctly disposing of waste from carton packaging and manufacturing ecological tiles. In other words, the transformation of material discarded, either pre- or post-consumer, into a viable final product depends on the integration between the actors of the entire ecosystem. At this point, the role of the final consumers of food products packaged in carton packaging stands out as decisively responsible for the correct destination of post-consumer packaging. As for barriers, B13, B14 and B15 highlight the difficulties associated with people, companies, and inspectors concerning environmentally sustainable practices, including the correct destination of the waste produced. Therefore, even visibly necessary cooperation among actors still faces obstacles.

As a third topic of discussion, the difficulty of obtaining pre- and post-consumer material for the manufacture of ecological tiles was imposed mainly after the beginning of the pandemic, as evidenced by the barriers of the "Lack of material" category. This difficulty is the result of failures in selective collection and in the collection of unqualified material (barriers B1, B3 and B5) and the reduction in the manufacture of carton packaging and in the disposal of pre-consumer packaging by the food industries (barriers B2 and B4).

While selective collection and the correct disposal of the material to be recycled are issues that can be improved, as can be seen by drivers D15, D16 and D17, the reductions in the manufacture of carton packaging and in the disposal of pre-consumer materials represent, on their own, an advance towards circularity. From a circular point of view, reduction is a more sustainable alternative when compared to recycling, since recycled materials need additional energy expenditure [71]. In this sense, manufacturing in smaller quantities or reducing industrial waste are extremely beneficial practices for the environment.

However, the reduction either in the manufacture of carton packaging or in the disposal of pre-consumer waste is a new reality to which ecosystem actors must adapt. This

change requires companies to think about new alternatives for acquiring material. In an attempt to adapt, ecosystem actors engaged in the study of the feasibility of new materials such as PET, a material that is difficult to recycle, to be mixed with the material necessary for the manufacture of ecological tiles. Thus, as the lack of material became an obstacle to be overcome, the search for new materials demonstrates, once again, an advance towards circularity. Expanding the application of previously unexplored materials is essential to ensure that waste is not incorrectly disposed of and sent to landfills and that increased value is attached to the various types of waste.

Despite the efforts of actors to adopt new materials for the manufacture of ecological tiles, the implementation process still takes a long time. As a result, one of the consequences that tile manufacturers are facing at the moment is the idleness of the machines. As the availability of the material is not enough to meet the current demand for ecological tiles, companies end up limiting the number of machines used.

Given the difficulty of manufacturing ecological tiles, the competition between ecological tiles and fiber-cement tiles, less sustainable options from the point of view of the material and the production process, becomes even stronger. The idleness of the machines in the factories of ecological tiles directly influences the productive capacity of this industry, which is already considerably smaller compared to the fiber-cement tile industry. As the main material used for the manufacture of these tiles is cement, manufacturers are able to acquire the necessary material with much greater ease than those of ecological tiles. This is one of the reasons why ecological tiles still have a very small market share in the Brazilian tile market of around 5%, according to interviewee 4. In this sense, it appears that stimulating the circularity of the ecosystem through the optimization of collected materials and a high level of productivity is essential for ecological tiles manufacturers to be able to maintain themselves in an extremely competitive market, still dominated by fiber-cement tiles.

Finally, the importance of the role of society in the circularity of the ecosystem is highlighted. At the beginning of the ecological tiles business, the manufacturers had difficulties in entering the tile market since the recycled products had a negative connotation of "low quality" and "non-durable". It is attributed to this period of low level of market acceptance, mainly the lack of awareness of the population about the importance of recycling and the opportunities arising from this process. In contrast to what happens in European countries, the low quality of Brazilian education reflects the lack of awareness of the population about sustainability and its benefits. This reality, however, has been progressively changing, as the topic is gaining more importance and space. Despite the increase in awareness, it is observed that the niche of consumers of ecological tiles is still very restricted to commercial establishments and farms. Today, it appears that, despite an increase in demand for ecological tiles, which contributes to the appreciation of recycled products, there is still a lot of space for growth in the consumer market.

Based on our findings and discussions, we proposed a new theoretical framework of drivers and barriers to the development of a circular ecosystem (Figure 4).

Figure 4 summarizes both the characterization of the ecosystem in the six spheres proposed by Rong et al. [58] and the results related to the drivers and barriers discussed. Through the representation, it is possible to briefly observe the elements that build the ecosystem identified separately within the context, construct, cooperation, configuration, capability and change categories. It is also possible to visualize the categories of drivers and barriers, which enhance and hinder, respectively, the functioning of the ecosystem, as indicated by the forward and backward arrows. Some of the points discussed can also be observed in the framework, such as the issue of the emergence of the ecosystem, intrinsically related to the category of context, and the difficulties of collecting post-consumer material, demonstrated in the category of change.

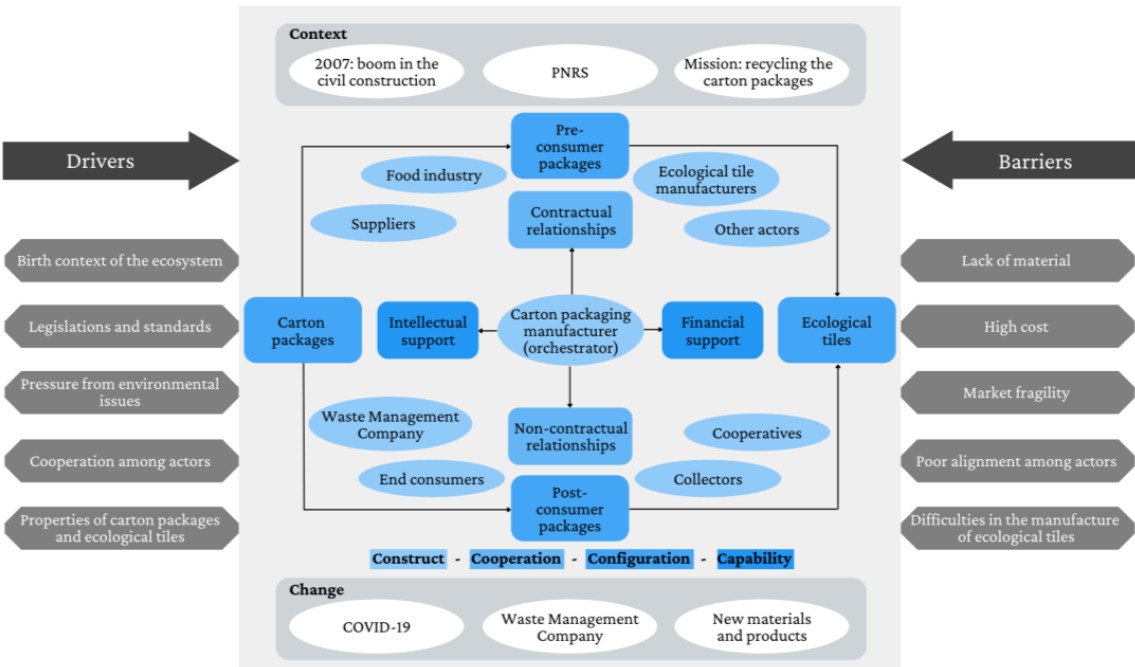

**Figure 4.** Theoretical framework.

## 6. Theoretical Contributions

This paper contributes to the circular economy transition and ecosystem literature [11,22,26,28,72] by analyzing an empirical study of a circular ecosystem, which is considered a recent and growing concept. We deeply explored twenty-seven drivers and seventeen barriers of an ecological tiles circular ecosystem set in the Brazilian context, where there are no records of articles in this area.

Although the current scholarship addresses the drivers and barriers of a circular economy at a micro-level [5,6,40] or the ecosystem bottlenecks [49,73], we elaborated on these theories and expanded the analysis by considering a circular ecosystem perspective. To date, no prior research proposed a theoretical framework that combines the drivers, barriers and current changes that the circular ecosystem faces. The framework shows the unique factors that enhance and hinder the circular ecosystem's existence and functioning.

Our theorization suggests that cooperation among actors can be critical to motivating the fragile ecosystem dynamic. Even though prior works demonstrate some ecosystem attributes, such as actors' alignments [22] and collaborations [21], these studies do not consider the circular value proposition within the ecosystem [26] and the sustainable aspects that surround the whole network. Our study reveals that legislation, information and material flow are much more critical in this type of configuration than compared to others.

Furthermore, the legislation has a great influence on the circularity within the ecosystem, even more than drivers associated with financial issues. In line with other research [14,28,30], our empirical findings emphasize that this type of structure has the potential to transform a linear production model into a circular one.

## 7. Practical Implications

Regarding the practical contribution, this work allows other companies to take advantage of these records for future insights regarding the formation of circular ecosystems, adapting them according to their realities. Our work reveals that changes within the ecosystem frequently occur (e.g., the reduction in the disposal of carton packages or the pressure to adopt sustainable solutions). The sooner companies realize that those changes must be seen as new opportunities, the more they can focus on new searchings and technologies de-

velopment, not only contributing to the survival of the ecosystem, but also to the expansion of its market and influence.

Other points that can be highlighted are the importance of establishing collaborations among actors and how those relationships occurred in the studied ecosystem. For that reason, this work might be particularly helpful for companies that still find it hard to establish partnerships and therefore are facing some troubles to maintain their competitiveness. Through the collaboration examples previously mentioned, those companies might pay closer attention to the external and internal factors (e.g., the enforcement of the National Policy on Solid Waste) that ultimately bring them to work together and understand the role of each actor to ensure the circular value proposition.

Besides, we emphasize the need to better educate and inform the Brazilian population about sustainable and circular opportunities. Awareness programs and local initiatives with communities, led by the public and the private sector, are extremely important to boost eco-awareness and, consequently, business models based on sustainability and circularity.

## 8. Conclusions

This study elucidates the functioning of a circular ecosystem, focused on the manufacture of ecological tiles from recycled carton packaging. Contrary to many empirical studies on business ecosystems that consider a single actor's restricted and exclusive view, this case study prioritized key actors' different points of view in the ecological tiles ecosystem. After conducting the interviews, including the one with the orchestrating company, it was possible to present a characterization of the ecosystem based on the 6C framework, originally developed by Rong et al. [58]. Furthermore, the drivers and barriers that enhance and restrict, respectively, the functioning of the studied circular ecosystem were presented and discussed.

It was found that, among several actors that cooperate with each other, the carton packaging manufacturer is seen as the orchestrator of the ecosystem. This is because it was the company responsible for the birth of the ecosystem, from the development of its mission to the provision of intellectual and financial support to the main actors involved—recyclers and ecological tiles manufacturers—to the development of the necessary processes. After the initial phase of the ecosystem, the orchestrator continued to invest in important partnerships and to generate value for its waste, pre- and post-consumer packaging. An example is the project signed between the orchestrator and the waste management company during the COVID-19 pandemic.

Regarding the drivers and barriers identified, some important points can be highlighted. The first one concerns the fundamental role of each actor within the ecosystem, especially the cooperative relationship established among them, which allowed to promote the delivered value proposition. This cooperation, however, proved to be poor in some aspects, given the lack of awareness of society and companies within the ecosystem, as mentioned above. Another highlight is the importance of the players being able to adapt to new realities, given the limited material availability. The ability to change often becomes crucial for the survival of the circular ecosystem.

This study has some limitations that can be converted into opportunities for future studies. First, the number of interviewees at each company, in which the primary data were obtained, was small. In addition, the limitations imposed by the single case study are that, despite enabling a deeper understanding of the object of study, it also prevents generalizations from being made. Considering that the focus of this work was on qualitative analysis, we strongly recommend that further studies focus on quantitative data regarding the financial viability and significance of recycling packages in Brazil. It is also possible to extend this study to different circular ecosystems in order to allow for more generalist analyses. Finally, it should be noted that continuous efforts are needed to identify new drivers and barriers within the context of circular ecosystems since they are constantly evolving.

**Author Contributions:** S.B., A.H.S., A.H.T. and J.M. were responsible for the conceptualization of the paper. The methodology was designed by A.H.S., A.H.T. and J.M. and written by A.H.S. and A.H.T. were in responsible for the data collection. S.B. and A.H.S. were responsible for the data analysis. The original draft was created by S.B. and A.H.S. The validation and formal analysis was realized by A.H.T., C.G.C., L.A.d.V.G. and J.M. The paper review and editing was done by all authors. All authors have read and agreed to the published version of the manuscript.

**Funding:** This research was funded by the São Paulo Research Foundation (FAPESP), under the processes 2021/03237-8, 2020/15831-9 and 2019/23655-9.

**Informed Consent Statement:** Informed consent was obtained from all subjects involved in the study.

**Acknowledgments:** The authors would like to thank The São Paulo Research Foundation (FAPESP), for financially supporting this research, under the processes 2021/03237-8, 2020/15831-9 and 2019/23655-9. The opinions, hypotheses, conclusions and recommendations expressed in this material are the responsibility of the authors and do not necessarily reflect the views of FAPESP. The authors also thank the support of Federal Institute of Education, Science, and Technology of Minas Gerais—Campus Congonhas. The authors are grateful for the valuable contribution and time dedicated by the interviewees that made this research possible.

**Conflicts of Interest:** The authors declare no conflict of interest.

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
