# Peer review of "Exploring the Dynamic of a Circular Ecosystem: A Case Study about Drivers and Barriers"

_sustainability, doi:10.3390/su14137875_

Round 1
Reviewer 1 Report
Comments to the Author (please copy-paste your report here if possible) Research on Exploring the Dynamic of a Circular Ecosystem: a Case Study about Drivers and Barriers. I commend the authors for conducting this review and on using a scientific, structured review process. I believe this paper adds important knowledge; however, I feel the current version is under-developed. There is an issue that need to be addressed. Below are my detailed comments for the current version.

Reviewer 2 Report
My comments and recommendations:
1. I would recommend to explain more in detail the Brazilian context of changes in the consumers' level of "eco-awareness" (in part Introduction), as well as to formulate proposals on how companies could help to improve it (in part Discussion, and Practical Implications).
2. The authors identify NGOs as one of the orchestrator partners - actors of the ecological tiles circular ecosystem (p. 8, Fig. 2). I expected that representatives of NGO/NGOs will be interviewed, too. Their perspective, responses to questions, and faced challenges could enable a broader and deeper understanding of the whole ecosystem.
3. Even though the extension of research findings and conclusions is limited (because of the qualitative character of the study) proposed new theoretical framework (p. 18, Fig. 4) enables authors to continue by defining quantifiable KPIs for individual drivers and barriers in further studies.
Reviewer 3 Report
researcher carried out a well good research work
Reviewer 4 Report
The work is part of a very new line of research that is attracting a lot of attention both in academia and in business management.
The revised literature appears consistent and fairly up-to-date.
However, the organization of the article structure should be revised. The first part of the introduction is set up as a literature review and lacks the contextualization of the research. On the contrary, the first part of the literature review on drivers and barriers (par 2.1) seems more introductory to the research context, highlighting better the gap that the authors want to fill. It is therefore more suitable for the introduction.
Regarding the methodology, some choices are not clear. The single case study method is selected but the choice of the case is in no way justified. Why was it chosen as a functional case for the purpose of the research?
Some doubts also arise regarding the different data collection procedure adopted for the 7 interviewees. Although triangulation was made of the sources of the data collected, the same sources were not used in all cases. Furthermore, in par. 3.1 it says first "The primary data came from interviews with 5 different companies", then "After that, as shown in Table 1, other 4 companies were interviewed", but in Table 1, only 7 respondents are reported. Some obviously belong to the same company, but the correspondence between the text and the table should be made clearer. There are no 9 companies analyzed.
In summary, the design of the research is confused in the initial part. It must be specified immediately that this is a qualitative research, justify the choice of the case study and declare the type of analysis (that it is an exploratory case study is only mentioned in the abstract).
Despite these doubts, the subsequent arguments relating to the results and discussion are handled very well. The theoretical implications are good. It is suggested to broaden the practical implications of the study to try to overcome the limit on the generalization of the results.
Congratulations and good luck
Round 2
Reviewer 4 Report
I believe that the small changes made as a result of my comments and those of the other reviewers have significantly improved the readability of the paper which previously left some small uncertainty. I renew my congratulations to the authors.